# Process Optimization of Microwave Assisted Simultaneous Distillation and Extraction from Siam cardamom using Response Surface Methodology

**Panawan Suttiarporn** [1], **Nalin Wongkattiya** [2], **Kittisak Buaban** [2], **Pisit Poolprasert** [3] **and Keerati Tanruean** [4,*]

1   Faculty of Science, Energy and Environment, King Mongkut's University of Technology North Bangkok, Rayong Campus, Rayong 21120, Thailand; panawan.s@sciee.kmutnb.ac.th
2   Program in Biotechnology, Faculty of Science, Maejo University, Chiang Mai 50290, Thailand; nalin.wongkattiya@gmail.com (N.W.); kittisak.cue@gmail.com (K.B.)
3   Program in Biology, Faculty of Science and Technology, Pibulsongkram Rajabhat University, Phitsanulok 65000, Thailand; poolprasert_p@psru.ac.th
4   Program in Biotechnology, Faculty of Science and Technology, Pibulsongkram Rajabhat University, Phitsanulok 65000, Thailand
*   Correspondence: keerati.t@psru.ac.th

**Abstract:** The main goal of the research was to optimize microwave-assisted simultaneous distillation and extraction (MA-SDE) using response surface methodology (RSM), based on Box–Behnken design (BBD). A process was designed to extract the essential oil from the leaf sheath of Siam cardamom. The experimental data were fitted to quadratic equations, and the experiment conditions for optimal extraction of 1,8-cineole were extraction time 87.68 min, material-to-water ratio 1:13.18 g/mL and microwave power 217.77 W. Under such conditions, the content of 1,8-cineole was 157.23 ± 4.23 µg/g, which matched with the predicted value. GC–MS results indicated the presence of predominant oxygenated monoterpenes including 1,8-cineole (20.63%), iso-carveol (14.30%), cis-*p*-mentha-1(7),8-dien-2-ol (12.27%) and trans-*p*-2,8-menthadien-1-ol (9.66%), and oxygenated contents were slightly higher in the MA-SDE and extraction compared to usual SDE. In addition, the essential oil extracted by MA-SDE exhibited strong antibacterial effects against the tested Gram-positive bacteria. Scanning electron micrographs provided more evidence of destruction of the leaf sheath treated by MA-SDE. Conclusively, microwave-assisted simultaneous distillation and solvent extraction appear to be an effective technique for the separation of essential oils enriched 1,8-cineole from Siam cardamom leaf sheath in a shorter time.

**Keywords:** microwave-assisted simultaneous distillation and extraction; leaf sheath of Siam cardamom; 1,8-cineole; response surface methodology

## 1. Introduction

*Wurfbainia vera* (Blackw.) Skornick. & A.D.Poulsen (synonym: *Amomum krervanh* Pierre ex Gagnep.) [1], known as Siam cardamom, is one of the most commonly found spices and plants in Southeast Asia that is also famous for its medicinal properties. It is an aromatic plant belonging to the family Zingiberaceae. The plant consists of several parts such as underground rhizomes; the pseudostem, which is made of leaf sheath; and fruit. The leafy shoots are formed by long, sheathlike stalks encircling one another. While, the cardamom seeds are often used as an optional ingredient in spice mixtures, the leaf sheath of Siam cardamom is also used as one of the ingredients in popular Eastern Thai dishes, such as wild boar with cardamom and boiled chicken with cardamom, due to

its specific fragrant aroma. The Siam cardamom gives a nice flavor. It is generally accepted by locals that it reduces dyspepsia and flatulence caused by indigestion. Essential oils (EOs) obtained from herb materials are composed mainly of terpene, oxygenated terpene and non-terpene components. Cardamom is the major source of 1,8-cineole, which is present in its essential oil. 1,8-cineole ($C_{10}H_{18}O$) is a nutraceutical monoterpenoid, also known as eucalyptol. 1,8-cineole also has therapeutic benefits such as antidiabetic and hypocholesterolemic properties [2], anti-inflammatory [3], antimicrobial activity [4], and antispasmodic effect [5]. Moreover, it is often used as a flavoring agent for food products and is also employed by the pharmaceutical industry in aromatherapy as a skin stimulant in the form of bath products for skin [6].

Different methods are used to isolate essential oils from various aromatic plants. Among them, hydro- or steam distillation has been the most effective technique used to extract the essential oils from medicinal plants. It is also widely accepted as a traditional method for essential oil extraction. However, these conventional methods have been limited due to several reasons, such as the relatively small amount of yield produced, time consumption and loss of some components [7,8]. To improve the extraction efficiency, simultaneous distillation and extraction (SDE) using the Likens–Nickerson apparatus meant the steam distillation extraction combined with an advantage associated with liquid–liquid extraction. The compounds are first distilled and then extracted into an organic solvent. SDE is quite useful for the extraction of the low content of volatile compounds, such that its extraction efficiency and recovery yield of essential oils was higher than other traditional techniques [9].

A new approach, microwave-assisted extraction (MAE), which is an auxiliary technique, has been used to enhance extraction performance while also reducing operating costs. This is mainly due to the microwave contributing to making heating more effective and selective while accelerating energy transfer and response to control the heat. It also helps to reduce thermals [10]. The utilization of microwave heating in the SDE extraction process was adopted as an essential alternative in green extraction techniques, namely, microwave-assisted hydrodistillation (MAHD). The interaction of microwave irradiation between extraction media, water molecules and polar compounds causes oscillation and heat up rapidly to generate the pressure within walls of cells. In addition, microwaves disrupt the structure of plant cell walls, promoting the release of oils into enclosing water, making the extraction process quicker [11]. MAHD has been performed for the extraction of essential oils from plants such as *Rosmarinus officinalis* L. [7,8], *Xylopia aromatic* [12], *Carum ajowan*, *Cuminum cyminum*, *Illicium anisatum* [13], *Thymus vulgaris* L. [14] and *Callicarpa cana* L. [15].

Considering the advantages of the simultaneous distillation and extraction (SDE) apparatus, the combination of SDE and microwave oven as a heating system, namely, microwave-assisted simultaneous distillation and extraction (MA-SDE), was adapted for the extraction of essential oils in order to yield concentrated analytes due to the continuous recycling of extraction solvent [16]. The microwave-assisted system is used to replace conventional heating sources which used to be time-consuming. Nevertheless, there have been no reports in regard to the extraction of essential oil of the cardamom leaf sheath utilizing the MA-SDE method.

In this research, an integrated microwave-assisted SDE apparatus is used to extract the essential oil out of the leaf sheath of *Wurfbainia vera*. A process of MA-SDE variables, including the extraction time, material-to-water ratio and microwave power, were optimized using response surface methodology (RSM) with a Box–Behnken design (BBD) based on the yield of 1,8-cineole. Moreover, the optimized MA-SDE process was compared to the conventional SDE technique used to produce the essential oils from the leaf sheath of *Wurfbainia vera* in terms of the content of 1,8-cineole, aromatic compositions and antibacterial activities, as well as microstructure change of plant materials.

## 2. Materials and Methods

### 2.1. Sample Preparation

The materials used in this experimental work were Siam cardamom (*Wurfbainia vera*), which was grown and harvested in Pongnamron district, Chantaburi province (12°53′28″ N, 10°17′30″ E). The plant

was authenticated by the Faculty of Science, Energy and Environment, the King Mongkut's University of Technology in North Bangkok, Rayong campus, Thailand. The plant sample was preserved as a specimen number (SEV-KMU-RY001-2019). After harvesting, the cardamom materials were washed with water to remove any impurities. The leaf sheath of the cardamom was dried at 40 °C in the oven to a point where constant weight was achieved before being ground and passed through sieves no. 35 (500 μm). The material powders were kept in a desiccator before being used.

### 2.2. Microwave-Assisted Simultaneous Distillation and Extraction (MA-SDE) Process

A schematic diagram of the MA-SDE apparatus is shown in Figure 1. The MA-SDE apparatus consisted of a microwave assisted extraction system, acting as the heat source, and a simultaneous distillation-extraction (SDE), modified Likens–Nickerson apparatus [17]. The household microwave oven (Electrolux, Bangkok, Thailand) was adapted in the laboratory. The maximum power used was 700 W, with the power source being 2450 MHz. The dimensions of the interior cavity of the microwave oven were 44.0 × 35.5 × 2.59 cm.

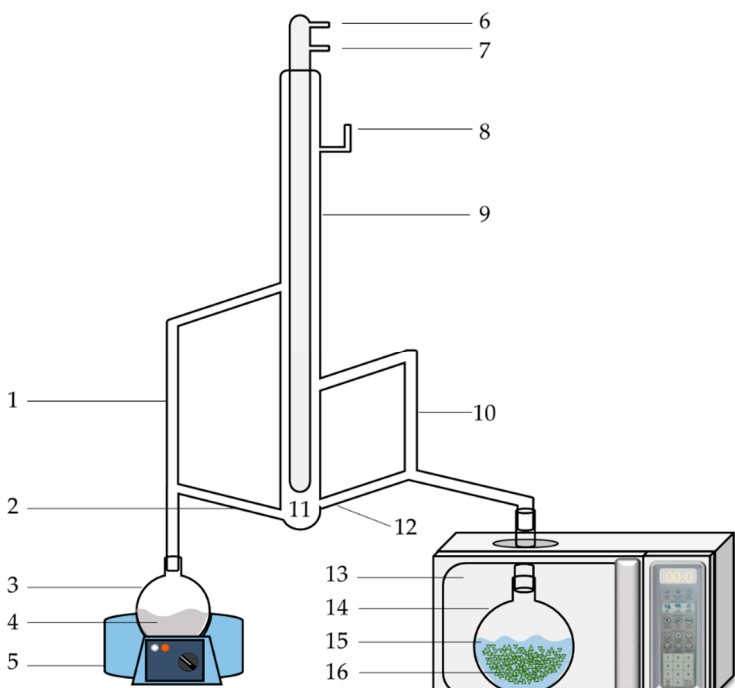

**Figure 1.** Schematic diagram of MA-SDE apparatus. (1) Vapor tube for organic phase; (2) return arm of organic phase; (3) flask I; (4) solvent; (5) heating mantle; (6) water outlet; (7) water inlet; (8) air-vent; (9) condenser; (10) vapor tube for aqueous phase; (11) organic and aqueous phase separation; (12) return arm of aqueous phase; (13) microwave oven; (14) flask II; (15) water; (16) plant material.

The extraction process of cardamom essential oil is shown in Figure 2. A 1000 mL flat-bottom flask II containing part of Siam cardamom sample and 300 mL distilled water were placed inside the microwave oven. A 100 mL aliquot of dichloromethane (Labscan, Bangkok, Thailand) was used as the organic phase placed in a 500 mL round-bottom flask I. The water in sample flask along with extracted organic solvent was heated to boil by microwave irradiation and the heating mantle, respectively. Then, the water vapor, with volatile organic compounds that were released from the plant cells and the organic solvent, was condensed in the condenser. The volatile compounds were partitioned from the aqueous into the organic phase. The organic solvent containing essential oil and water were then returned to each flask through return arms. After the extraction period, the dichloromethane extract was collected and dehydrated with $Na_2SO_4$ anhydrous (Sigma-Aldrich, St. Louis, MO, USA).

The extracted solution was then concentrated to about 1 mL at 40 °C with a rotary evaporator (Heidolph, Schwabach, Germany). The essential oil obtained was then stored at 2 °C for the GC–MS analysis.

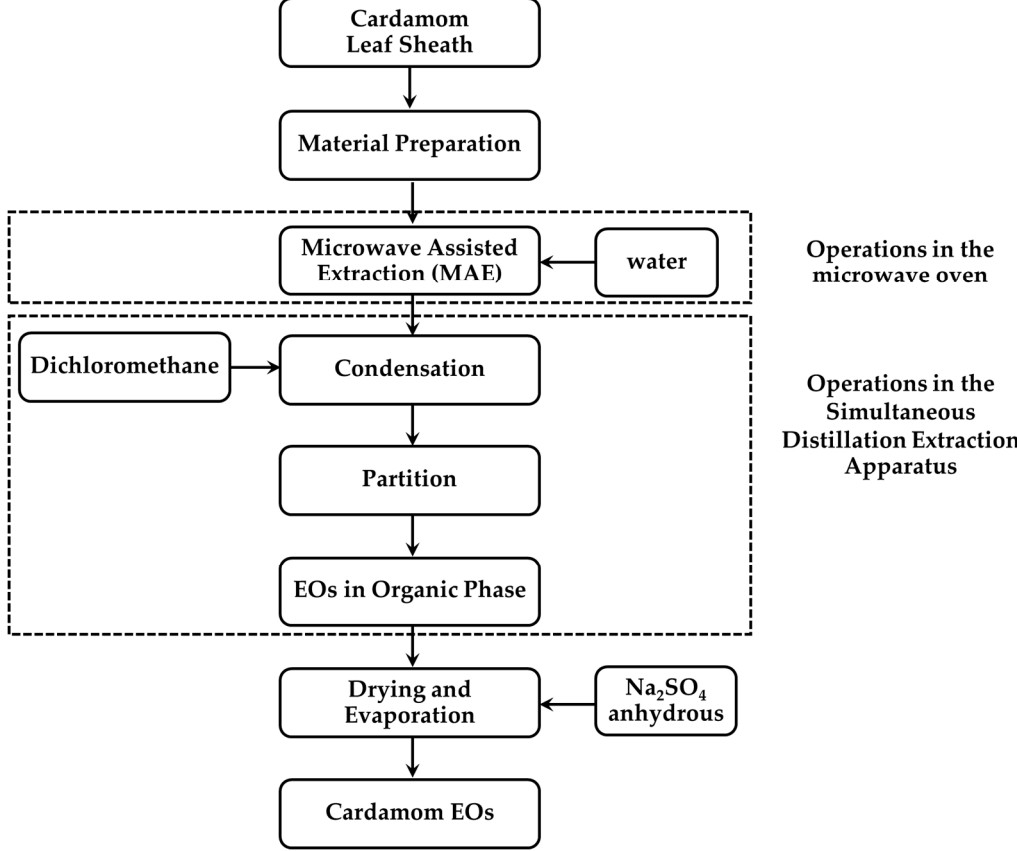

**Figure 2.** The process of essential oil extraction.

### 2.3. Conventional Simultaneous Distillation and Extraction

SDE was performed in a similar procedure as the one explained for MA-SDE. However, an electromantle (Mtops, Seoul, Korea, 500 W) was used as the heating source and the operation was carried out for 4 h according to the previous research [13] with a slight modification.

### 2.4. Gas chromatography–Mass Spectrometry (GC–MS) Analysis

GC–MS analysis of the chemical composition of essential oils was performed using an Agilent 7890B chromatograph, coupled with Agilent 5977B mass spectrometer (Agilent Technology, Santa Clara, CA, USA), equipped with an HP-5 MS capillary column (5% phenyl methyl siloxane, 30 m × 0.25 mm; 0.25 μm; film thickness). The electron impact ionization mode of the mass selective detector was operated with a mass scan range from *m/z* 29 to 550 at 70 eV ionization energy. The oven temperature was programmed to increase from 40 to 250 °C at a rate of 3 °C per minute. The injector temperature was set at 230 °C, and the detector temperature was 250 °C. Helium was the carrier gas utilized for this experiment, at a flow rate of 1 mL/min and a split ratio of 1:100. The area percentages of the volatile compounds were carried out by a peak area normalization measurement. The assigned compounds were identified by matching experimental mass spectra with those in the NIST mass spectral database. To confirm identification, retention indices were also investigated for all compounds by injecting a homologous series of n-alkanes ($C_9$–$C_{26}$) under the Kovats condition and comparing the relative retention indices with the values given in the literature. The external standard method was used to quantify the major constituents in the Siam cardamom essential oil. The essential oil samples and 1,8-cineole standard (10–500 ppm) were diluted using dichloromethane. The correlation coefficients

($R^2$) of the five-point calibration curves obtained were 0.9996. The contents of 1,8-cineole (µg/g) in the essential oils were determined using the quantitative data based on the standard curves.

## 2.5. Single Factor Design

A single factor design was performed to select an appropriate range of extraction factors. The parameters, including extraction time (40, 80 and 120 min), material-to-water ratio (1:10, 1:12, 1:15, 1:20 and 1:60 g/mL) and microwave power (70, 140, 210, 280 and 350 W), were all investigated. A single factor experiment was used to study the effect of different factors on the response, including the extraction yield and contents of 1,8-cineole. Each trial was analyzed in triplicate and the standard deviation was calculated. The extraction yield was determined using the following equation:

$$\text{Yield of essential oil } (\%) = \frac{\text{Mass of extracted essential oil}}{\text{Mass of dried material}} \times 100 \tag{1}$$

## 2.6. Optimization of the MA-SDE Process Using RSM Procedure

Response Surface Methodology (RSM), in combination with Box–Behnken design (BBD), was selected to construct a statistic model for the microwave-assisted simultaneous distillation and extraction (MA-SDE) process. The suitable ranges of extraction time ($X_1$), material to water ratio ($X_2$) and microwave power ($X_3$) for the extraction of Siam cardamom essential oil-rich 1,8-cineole were obtained based on a single factor test, as shown in Table 1. Each independent variable consisted of 3 levels ranging from low (−1), to medium (0), and, to high (+1). The three-levels–three-factors BBD design was employed to determine the content of 1,8-cineole using Minitab statistic software (Trial version 17, Minitab Inc., State College, PA, USA). The complete design consisted of 15 experiments with 3 center points. The significance of the model equations and the model terms were carefully evaluated in terms of *p* values using an analysis of variance (ANOVA) with a 95% confidence level. This model can be expressed with the coded variables ($X_1$, $X_2$, and $X_3$) using the following equation:

$$Y = \beta_0 + \sum_{i=1}^{3} \beta_i X_i + \sum_{i=1}^{2} \beta_{ii} X_i^2 + \sum_{i=1}^{2} \sum_{j=i+1}^{3} \beta_{ij} X_i X_j + \varepsilon \tag{2}$$

where *Y* is the 1,8-cineole dependent response, $X_i$ is the independent parameter, $\beta_0$, $\beta_i$, $\beta_{ii}$ and $\beta_{ij}$ are the regression coefficients of the intercept, linear, quadratic and interaction terms, and "$\varepsilon$" is the random error.

**Table 1.** Coding of independent variables and factor levels.

| Independent Variables | Symbol | Factor Level | | |
|---|---|---|---|---|
| | | **−1** | **0** | **1** |
| Extraction time (min) | $X_1$ | 40 | 80 | 120 |
| Material-to-water ratio (g/mL) | $X_2$ | 1:10 | 1:12.5 | 1:15 |
| Microwave power (W) | $X_3$ | 140 | 210 | 280 |

The model fitting was evaluated based on R-squared, adj-R-squared, predicted-R-squared, and the lack-of-fitness statistic. Optimal conditions for MA-SDE of the 1,8-cineole compound from cardamom depended on factors such as extraction time, material to water ratio and microwave power, which were obtained using the predictive equations of RSM. The response surface plots were developed using the Statistica program (Trial version 10.0, Statsoft Inc., Tulsa, OK, USA).

## 2.7. Antibacterial Activity Testing

Bacterial strains (Staphylococcus aureus DMST 8840, Streptococcus pyogenes DMST 30563, Bacillus cereus DMST 5040, Listeria monocytogenes DMST 17303, Escherichia coli DMST 4212, Salmonella

Typhi DMST 5784, Pseudomonas aeruginosa DMST 4739 and Enterobacter aerogenes DMST 8841) were obtained from Department of Medical Science of Thailand, Ministry of Public Health, in Thailand. They were subcultured onto Brain Heart Infusion Agar (BHA) at 37 °C for a duration of 24 h prior to testing. The bacterial suspension was adjusted to match the turbidity standard of 0.5 McFarland. Each one of the bacterial suspensions was spread onto Mueller Hinton (MH) Agar using a sterile cotton swab. Ten microliters of the sample were dropped on a 6 mm diameter disc (Macherey-Nagel, Düren, Germany). The disc was placed onto the inoculated BHA and incubated at 37 °C for 24 h before the inhibition zone was observed. Tetracycline (Oxoid, Hants, UK) was used as a control. This assay was performed in triplicate.

### 2.8. Scanning Electron Microscopy (SEM)

After the extraction of the essential oil by MA-SDE and SDE processes, any changes in the microstructure of cardamom leaf sheath were examined by SEM analysis. The micrographs of the treated samples were compared to the untreated samples. At this stage, the samples were fixed on an aluminum sample holder and then sputtered with gold in a sputter coater. All of the samples were investigated using electron microscope scanning (JSM-IT500HR, JEOL Ltd., Tokyo, Japan) at an accelerating voltage of 10.0 kV.

### 2.9. Statistical Analysis

The results were expressed as mean ± from standard deviation. Statistical analysis was through the paired *t*-test ($p < 0.05$) and ANOVA which utilized Duncan's multiple comparison test to determine significant differences ($p < 0.05$) identified among the means using SPSS (version 26, IBM Institute Inc., Endicott, NY, USA).

## 3. Results and Discussion

### 3.1. Composition Identification Using GC–MS

The components of Siam cardamom leaf sheath essential oil obtained by MA-SDE and SDE were identified by GC–MS according to retention index and NIST (Table 2). The total ion chromatograms (TIC) were shown in Figure 3. The percentage of relative abundance was calculated using the peak area normalization method, which is represented as the percentage of their peak areas in total peak areas. As shown in Table 2, a total of 24 components were detected, among which MA-SDE essential oil consisted of 24 components and SDE essential oil of 21 components, respectively. The twenty four identified compounds were classified, which included fifteen oxygenated monoterpenes (1,8-cineole, iso-carveol, cis-*p*-mentha-1(7),8-dien-2-ol, etc.); three sesquiterpenes (humulene, *β*-selinene and caryophyllene); and six oxygenated sesquiterpenes (humulene epoxide II, caryophyllene oxide, humulenol, etc.). The detected components represent 99.95% and 98.70% of the essential oils obtained by MA-SDE and SDE. The volatile compositions of essential oil obtained using both extraction methods were very similar, but the percentage of relative abundance difference was observed. The main compounds found in the essential oil obtained by MA-SDE and SDE were 1,8-cineole, iso-carveol, cis-*p*-mentha-1(7),8-dien-2-ol and trans-*p*-2,8-menthadien-1-ol. This result, which indicates that essential oil could be extracted using MA-SDE without causing chemical profile changes in essential oil components, is consistent with previous reports [18,19]. 1,8-Cineole, which was the major constituent in the essential oil reported in this study similarly was also the most abundant when compared to the report on the essential oils of cardamom fruits and seeds collected from other regions, as shown in Table 3. [4,20,21]. However, the other compounds of leaf sheath cardamom oil were different from the essential oil of cardamom seed that exhibited 1,8-cineole (70.87%), *α*-pinene (8.89%) and limonene (4.81%) [20]. In comparison to the conventional SDE method, it is considered that the use of MA-SDE had an impact on essential oil yield and the content of the main oxygenated compound, 1,8-cineole, with a significant difference ($p < 0.05$). Moreover, the essential oil acquired by MA-SDE

(96.75%) had a slightly higher content of oxygenated compounds than that obtained from SDE (95.17%) with significant different ($p < 0.05$). Most components in Siam cardamom essential oils were oxygenated compounds, which have a greater dipole moment and response to microwave irradiation compared to hydrocarbons. [22]. Therefore, it was easier to isolate the Siam cardamom essential oil from the material through microwave-assisted extraction.

**Table 2.** Identified compositions of essential oil from Siam cardamom by GC–MS analysis.

| Peak | RI | Compounds | Molecular Formula | % Relative Abundance | |
|------|-----|-----------|-------------------|--------------|--------------|
| | | | | **MA-SDE** | **SDE** |
| 1 | 1033 | 1,8-cineole | $C_{10}H_{18}O$ | 20.63 ± 0.03 * | 18.54 ± 0.51 * |
| 2 | 1123 | *trans-p*-2,8-Menthadien-1-ol | $C_{10}H_{16}O$ | 9.66 ± 0.02 * | 9.20 ± 0.17 * |
| 3 | 1137 | *cis-p*-Mentha-2,8-dien-1-ol | $C_{10}H_{16}O$ | 6.50 ± 0.02 | 6.57 ± 0.08 |
| 4 | 1140 | L-Pinocarveol | $C_{10}H_{16}O$ | 1.83 ± 0.10 * | 2.26 ± 0.25 * |
| 5 | 1144 | (+)-Camphor | $C_{10}H_{16}O$ | 0.87 ± 0.03 * | 1.07 ± 0.01 * |
| 6 | 1162 | Pinocarvone | $C_{10}H_{14}O$ | 1.15 ± 0.05 | 1.20 ± 0.10 |
| 7 | 1166 | δ-Terpineol | $C_{10}H_{18}O$ | 1.08 ± 0.02 * | 1.33 ± 0.08 * |
| 8 | 1176 | Terpinen-4-ol | $C_{10}H_{18}O$ | 0.91 ± 0.06 * | 1.20 ± 0.09 * |
| 9 | 1193 | *Iso*-carveol | $C_{10}H_{16}O$ | 14.3 ± 0.01 * | 15.42 ± 0.12 * |
| 10 | 1194 | α-Terpineol | $C_{10}H_{18}O$ | 0.91 ± 0.01 | |
| 11 | 1198 | Myrtenol | $C_{10}H_{16}O$ | 0.84 ± 0.01 * | 0.96 ± 0.02 * |
| 12 | 1203 | (-)-*trans*-Isopiperitenol | $C_{10}H_{16}O$ | 3.16 ± 0.08 * | 5.40 ± 0.23 * |
| 13 | 1221 | *cis*-Carveol | $C_{10}H_{16}O$ | 5.87 ± 0.07 | 6.09 ± 0.14 |
| 14 | 1233 | *cis-p*-mentha-1(7),8-dien-2-ol | $C_{10}H_{16}O$ | 12.27 ± 0.06 * | 13.95 ± 0.23 * |
| 15 | 1246 | (-)-Carvone | $C_{10}H_{14}O$ | 2.51 ± 0.02 * | 2.73 ± 0.06 * |
| 16 | 1418 | β-Caryophyllene | $C_{15}H_{24}$ | 0.79 ± 0.08 * | 1.34 ± 0.04 * |
| 17 | 1452 | Humulene | $C_{15}H_{24}$ | 1.16 ± 0.05 | 1.09 ± 0.07 |
| 18 | 1485 | β-Selinene | $C_{15}H_{24}$ | 1.23 ± 0.10 | 1.11 ± 0.11 |
| 19 | 1565 | Nerolidol | $C_{15}H_{26}O$ | 0.95 ± 0.01 | |
| 20 | 1582 | Caryophyllene oxide | $C_{15}H_{24}O$ | 2.57 ± 0.05 * | 2.17 ± 0.18 * |
| 21 | 1609 | Humulene epoxide II | $C_{15}H_{24}O$ | 5.17 ± 0.05 * | 4.13 ± 0.09 * |
| 22 | 1632 | Humulenol | $C_{15}H_{24}O$ | 1.88 ± *0.04 | 1.22 ± 0.12 * |
| 23 | 1636 | 11,11-Dimethyl-4,8-dimethylenebicyclo[7.2.0]undecan-3-ol | $C_{15}H_{24}O$ | 2.70 ± 0.07 * | 1.72 ± 0.03 * |
| 24 | 1838 | Corymbolone | $C_{15}H_{24}O_2$ | 1.01 ± 0.05 | |
| Total identified compounds | | | | 99.95 ± 0.03 * | 98.70 ± 0.36 * |
| Oxygenated monoterpene | | | | 82.49 ± 0.11 * | 85.92 ± 0.24 * |
| Oxygenated sesquiterpene | | | | 14.26 ± 0.08 * | 9.24 ± 0.07 * |
| Total oxygenated compounds | | | | 96.75 ± 0.19 * | 95.17 ± 0.16 * |
| Sesquiterpene | | | | 3.18 ± 0.20 | 3.54 ± 0.21 |
| Total extraction time (min) | | | | 87.68 | 240 |
| Yield (%) | | | | 2.120 ± 0.25 * | 1.584 ± 0.21 * |
| 1,8-cineole content (µg/g) | | | | 157.23 ± 4.23 | 119.94 ± 5.65 |

Mean followed by * are statistically different (student's *t*-test; $p < 0.05$).

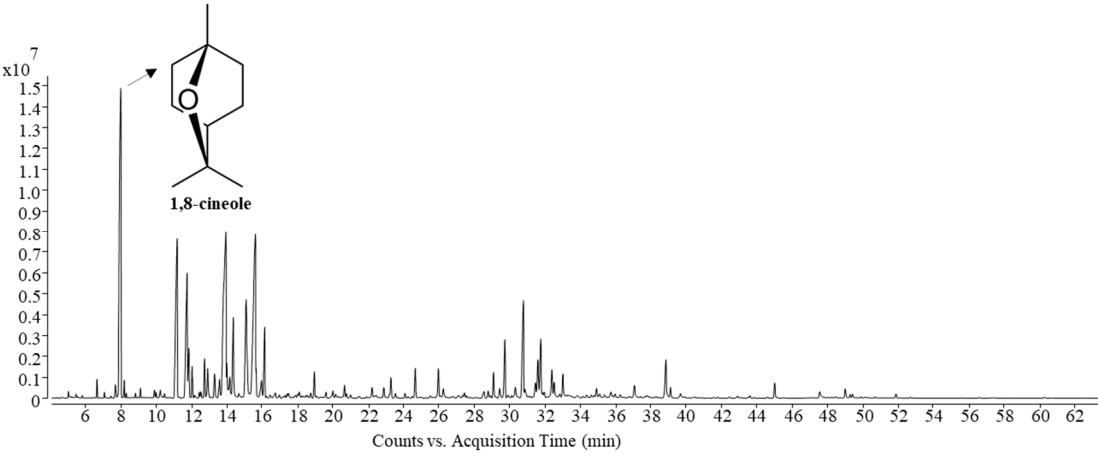

**Figure 3.** Total ion compound (TIC) chromatograms of the Siam cardamom essential oil obtained via MA-SDE.

**Table 3.** Major constituents of extracted essential oil of several studies relating to cardamom oil extraction.

| Origin | Cardamom Part | Constituent | Content (%) | Author |
|---|---|---|---|---|
| Thailand *Wurfbainia vera* (*Amomum krervanh*) | leaf sheath | 1,8-cineole isocarveol *cis-p*-mentha-1(7),8-dien-2-ol *trans-p*-2,8-menthadien-1-ol | 20.63 14.30 12.27 9.66 | This study |
| Thailand Reported as: *Amomum krevanh* | seed | 1,8-cineole *α*-pinene limonene | 70.87 8.89 4.81 | Yothipitak et al. (2009) [20] |
| China Reported as: *Amomum kravanh* | fruit | 1,8-cineole *α*-pinene *α*-terpinene *β*-pinene | 68.42 5.71 2.63 2.41 | Diao et al. (2014) [4] |
| China Reported as: *Amomum kravanh* | fruit | 1,8-cineole *α*-terpinyl acetate *β*-pinene | 59.7 5.0 3.1 | Feng et al. (2011) [21] |

### 3.2. Single Factor Investigation of the 1,8-cineole Compound

In order to optimize the essential oil extraction process by using the microwave, it is necessary to have an overall understanding of the factors that will influence the process. These factors include extraction time, microwave power and solid to liquid ratio, all of which significantly influence microwave-assisted extraction [23].

To achieve an appropriate result, the finding of optimal parameters and conditions is vital. The factors that affect the content of 1,8 cineole in the MA-SDE process are shown in Figure 4. Figure 4a showed the effect of extraction time on the amount of 1,8 cineole in essential oil. When extraction time was increased from 40 to 80 min, the amount of the compound in the extracted essential oil increased from 34.12 to 112.45 µg/g. Still, it decreased to 45.83 µg/g at 120 min due to degradation or loss of the volatile constituent caused by prolonging the process [24]. Hence, 40–120 min was the appropriate range for subsequent optimization by BBD.

To evaluate the effects of the material-to-water ratio, a series of extractions were carried out with different material-to-water ratios (1:10, 1:12, 1:15, 1:20 and 1:60 g/mL). As shown in Figure 4b, the extraction efficiency increased obviously with an increase of water before the material-to-water ratio reached 1:12 g/mL. It is essential to select the appropriate range of the material-to-water ratio in the distillation system because water vapor that is absorbed into the epidermis also contains essential oils, and volatile content can be affected and vaporized by the passing stream. Due to the presence of an excess amount of water, thermal stress occurs caused by the rapid heating of the solution on account of the effective absorption of microwaves by water [25]. On the other hand, an insufficient amount of water is unable to melt adhesives, the essential oils are not being released and target extraction is incomplete [26]. Therefore, 1:10–1:15 was chosen as a suitable material-to-water range for further optimization.

The effect of microwave power, as shown in Figure 4c, showed that by increasing the microwave irradiation power (from 70 to 210 W), it directly increases the 1,8 cineole content, while a further increase of microwave power to over 210 W resulted in a decrease in the compound content. The microwave irradiation power influences the efficiency of energy transfer onto the plant material. When more electromagnetic energy is transferred to the sample, an improvement of the extraction efficiency is observed, likely due to (1) providing a localized heating effect that lowers the viscosity and surface tension of the water and plant materials [27], and (2) direct heating water in the cells, exerting internal pressure and rupturing oil glands [28]. However, high microwave power may result in poor extract quality, which is attributed to the degradation of thermally labile compounds [29]. Hence, microwave irradiation power in the range of 140–280 W was selected for subsequent optimization.

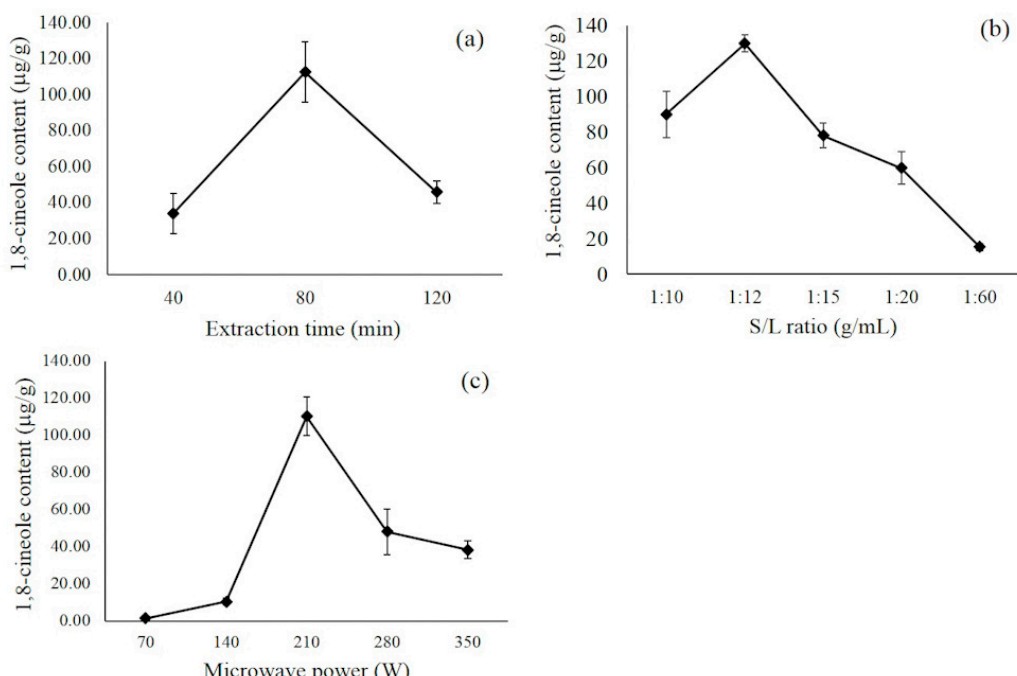

**Figure 4.** The effect of extraction time (**a**), material-to-water ratio (**b**), and microwave power (**c**) on 1,8 cineole content.

### 3.3. Optimization of Experimental Conditions on the Content of 1,8-Cineole Using RSM

Based on the single factor experiment, the acceptable range of conditions needed for extraction of the essential oil-rich 1,8-cineole has these three variables, including extraction time ($X_1$), material-to-water ratio ($X_2$) and microwave power ($X_3$), which were selected for further optimization using RSM. Following that, a three-levels–three-factors BBD was employed to construct 15 experimental designs for RSM randomly; the content of 1,8-cineole under different extraction conditions is shown in Table 4. The response variable and test variable can be related using the second-order polynomial equation as follows:

$$Y = 148.01 + 17.59X_1 + 31.54X_2 + 8.49X_3 - 43.08X_1^2 - 60.74X_2^2$$
$$- 46.33X_3^2 + 0.94X_1X_2 - 8.05X_1X_3 + 12.07X_2X_3 \tag{3}$$

The analysis of variance (ANOVA) that is shown in Table 5 was performed to investigate the significance and adequacy of the model terms. The suitability of the model was determined through statistic indicators, including *p*-values of the variables and lack of fit, and $R^2$. Concerning variable *p*-values, the suitability of this model was significant due to the low *p*-value ($p < 0.05$).

Meanwhile, the *p*-value of lack-of-fit was not significant ($p > 0.05$) relative to pure error, indicating that the quadratic model was statistically significant for the response. The high value of the determination coefficient ($R^2 = 0.9746$) and the adjusted determination coefficient (Adj $R^2 = 0.9288$) verified that the model was substantial and reasonable. Moreover, the linear term of extraction time ($X_1$) and material-to-water ratio ($X_2$), the quadratic term of extraction time ($X_1^2$), material-to-water ratio ($X_2^2$) and microwave power ($X_3^2$) were significantly different. In contrast, the linear term of microwave power ($X_3$) and all of the interaction terms among the independent variables were not significant.

**Table 4.** Design program and results of response surface.

| StdOrder | RunOrder | $X_1$ | $X_2$ | $X_3$ | 1,8 cineole Content (µg/g) |
|---|---|---|---|---|---|
| 15 | 1 | 80 | 1:12.5 | 210 | 150.36 |
| 10 | 2 | 80 | 1:15 | 140 | 45.77 |
| 8 | 3 | 120 | 1:12.5 | 280 | 83.12 |
| 2 | 4 | 120 | 1:10 | 210 | 16.37 |
| 11 | 5 | 80 | 1:10 | 280 | 12.13 |
| 14 | 6 | 80 | 1:12.5 | 210 | 141.41 |
| 7 | 7 | 40 | 1:12.5 | 280 | 45.13 |
| 1 | 8 | 40 | 1:10 | 210 | 1.97 |
| 12 | 9 | 80 | 1:15 | 280 | 92.38 |
| 9 | 10 | 80 | 1:10 | 140 | 13.51 |
| 6 | 11 | 120 | 1:12.5 | 140 | 88.03 |
| 3 | 12 | 40 | 1:15 | 210 | 70.15 |
| 5 | 13 | 40 | 1:12.5 | 140 | 17.84 |
| 4 | 14 | 120 | 1:15 | 210 | 88.03 |
| 13 | 15 | 80 | 1:12.5 | 210 | 152.68 |

**Table 5.** ANOVA for determining the fitting of the quadratic model.

| Source | Sum of Squares | Df | Mean Square | F-Value | *p*-Value | Remarks |
|---|---|---|---|---|---|---|
| Model | 36630.8 | 9 | 4070.1 | 21.29 | 0.002 | significant |
| $X_1$ | 2475.6 | 1 | 2475.6 | 12.95 | 0.016 | significant |
| $X_2$ | 7958.2 | 1 | 7958.2 | 41.63 | 0.001 | significant |
| $X_3$ | 576.5 | 1 | 576.5 | 3.02 | 0.143 | not significant |
| $X_1^2$ | 6852.2 | 1 | 6852.2 | 35.85 | 0.002 | significant |
| $X_2^2$ | 13620.7 | 1 | 13620.7 | 71.26 | 0.000 | significant |
| $X_3^2$ | 7950.8 | 1 | 7950.8 | 41.59 | 0.001 | significant |
| $X_1X_2$ | 3.5 | 1 | 3.5 | 0.02 | 0.897 | not significant |
| $X_1X_3$ | 259.2 | 1 | 259.2 | 1.36 | 0.297 | not significant |
| $X_2X_3$ | 583.0 | 1 | 583 | 3.05 | 0.141 | not significant |
| Lack-of-fit | 879.3 | 3 | 293.1 | 7.66 | 0.118 | not significant |
| Pure error | 76.5 | 2 | 38.2 | | | |
| Total | 37586.5 | 14 | | | | |
| $R^2$ 0.9746 | | | Adjusted $R^2$ 0.9288 | | Predicted $R^2$ 0.6211 | |

3D response surface plots and contour plots were constructed to illustrate the influence of mutual interaction of the independent variables on the MA-SDE procedure. Figure 5a–c shows the effects of extraction time, material-to-water ratio and microwave power on the MA-SDE process when it comes to the extraction of the Siam cardamom essential oil-rich 1,8-cineole. Visually, the plot indicated the presence of the maximum content of 1,8-cineole. The response increased proportionally as pairwise conditions increased. Until these conditions exceed the optimal point (87.68 min, 1:13.18 g/mL, and 217.77 W), the amount of obtained essential oil-rich 1,8-cineole increased and declined after that as the conditions kept increasing. The predicted content of 1,8-cineole by RSM under the optimal conditions with slight modification (87.68 min, 1:13.18 g/mL, and 210 W) was 153.93 µg/g. When verification experiments were carried out under optimal conditions, the actual response was 157.23 ± 4.23 µg/g. Based on the archived results of the actual experiments, they were not significantly different from those initially predicted by the models; it suggests that the optimal conditions were reliable.

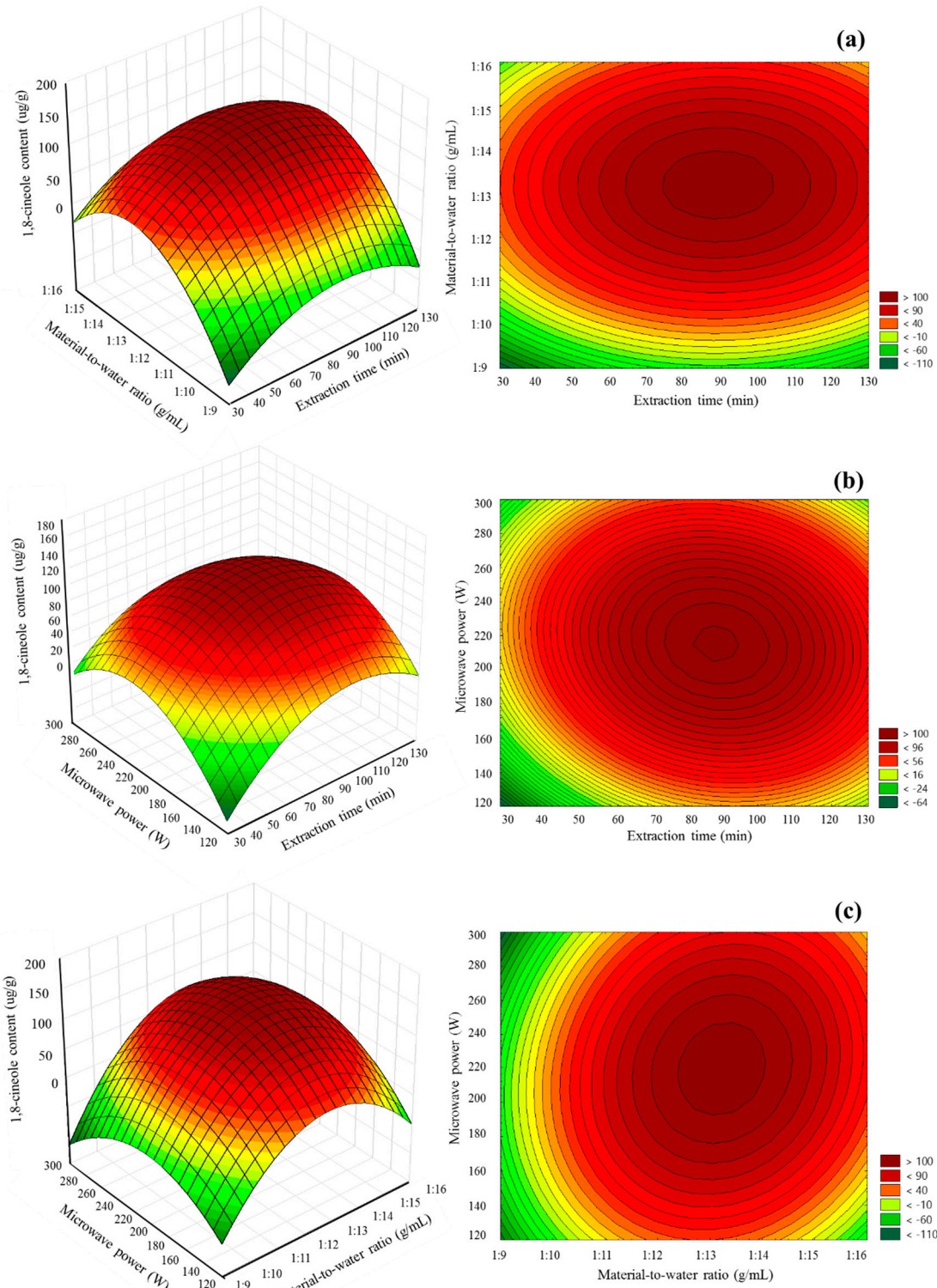

**Figure 5.** Response surface and contour plots: (**a**) effect of extraction time and material-to-water ratio; (**b**) effect of extraction time and microwave power; and (**c**) impact of material-to-water ratio and microwave power.

*3.4. Antibacterial Assay*

Antibacterial evaluation of essentials oils obtained by MA-SDE, SDE and 1,8-cineole standard was performed by utilizing agar disc diffusion assay. The results are shown in Table 6. The essential oil

obtained using both extraction techniques, including MA-SDE and SDE, showed strong antibacterial effects against the tested Gram-positive bacteria with ranges from 12–30 mm of the inhibition zone. On the other hand, they displayed lower inhibition against all tested Gram-negative bacteria (NA-11 mm.). The results are consistent with the previous studies conducted that concentrated on the sensitivities of specific foodborne pathogens to the essential oil [4]. In contrast, only pure 1,8 cineole showed slight antibacterial activity against both Gram-positive and Gram-negative bacteria. The antibacterial effect of crude oil (MA-SDE and SDE) was stronger than the pure compound (1,8 cineole). This could be attributed to the synergism effect of various compounds in crude oil. MA-SDE and SDE extracted essential oil showed slightly lesser antibacterial potency against Gram-positive bacteria than tetracycline. In contrast, MA-SDE extracted essential oil exhibited a greater antibacterial activity on *Str. pyogenes* DMST 30563, with 30.77 mm of the inhibition zone than the commercial antibiotics and SDE. This finding showed that MA-SDE was an appropriate method to obtain the essential oil which provides medicinal benefits. Therefore, the essential oil—mixtures of natural terpenes containing 1,8-cineole as a major—showed synergistic effects. Siam cardamom essential oils can be used as a nutraceutical or adjuvant to antibiotic therapy.

**Table 6.** Agar disc diffusion of essentials oils obtained by MA-SDE and SDE, and 1,8-cineole standard.

| Bacteria | Inhibition Zone (mm) | | | |
|---|---|---|---|---|
| | Tetracycline | MA-SDE | SDE | 1,8-cineole |
| *S. aureus* DMST 8840 | 24.89 ± 0.61 [a] | 16.99 ± 0.63 [c] | 19.00 ± 0.29 [b] | 6.78 ± 0.21 [d] |
| *Str. pyogenes* DMST 30563 | 19.97 ± 0.36 [b] | 30.77 ± 3.46 [a] | 18.92 ± 0.21 [b] | 9.91 ± 0.35 [c] |
| *B. cereus* DMST 5040 | 19.17 ± 0.33 [a] | 16.47 ± 1.68 [b] | 16.25 ± 0.35 [b] | 6.67 ± 0.58 [c] |
| *L. monocytogenes* DMST 17303 | 24.95 ± 0.85 [a] | 13.36 ± 1.92 [b] | 12.82 ± 1.37 [b] | 7.16 ± 0.29 [c] |
| *E. coli* DMST 4212 | 22.48 ± 0.67 [a] | 8.21 ± 0.37 [c] | 9.91 ± 0.28 [b] | 7.17 ± 0.29 [d] |
| *Sal.* Typhi DMST 5784 | 22.68 ± 0.40 [a] | 9.41 ± 0.20 [c] | 11.02 ± 0.36 [b] | 8.87 ± 0.06 [c] |
| *P. aeruginosa* DMST 4739 | 7.83 ± 0.26 | NA | NA | NA |
| *Ent. aerogenes* DMST 8841 | 18.81 ± 0.29 [a] | 8.61 ± 0.28 [b] | 8.75 ± 0.28 [b] | 8.33 ± 0.26 [b] |

NA: not applicable. The same letter in each row indicate that the means are not significantly different ($p < 0.05$).

### 3.5. Structural Changes

In order to observe any structural changes in the material after extraction processes using different extraction methods, SEM analysis was performed. The micrographs of untreated and treated materials are shown in Figure 6. As shown in Figure 6a, the external surface of the untreated plant's leafy shoot was smooth. After extraction by MA-SDE (87.68 min) (Figure 6b), most of the cell walls were destroyed by the microwaves to such an extent that large perforation had occurred on the leaf sheath surface. However, after the extraction was performed by SDE (240 min) (Figure 6c), the leaf sheath appeared wrinkled with only a few ruptures present.

In the MA-SDE process, the heating occurs based on microwave heating, which is, in contrast, to work on a different principle compared to the conventional SDE system. In the conventional SDE system, the heat transfer occurs via transfer from the outside to the inside of the plant where the temperature increases slowly; this process is highly dependent on the thermal conductivity and convection currents. Comparatively, the mass transfer occurs from the inside to the outside of the plant material. However, with the MAE system, the rapid rise in temperature and internal pressure by the MAE process occurred due to the microwaves selectively heating the intracellular polar molecules, resulting in ruptures promoting the effective release of substances from the cells. Both the heat and mass transfers were in the same direction from the inside to the outside [30].

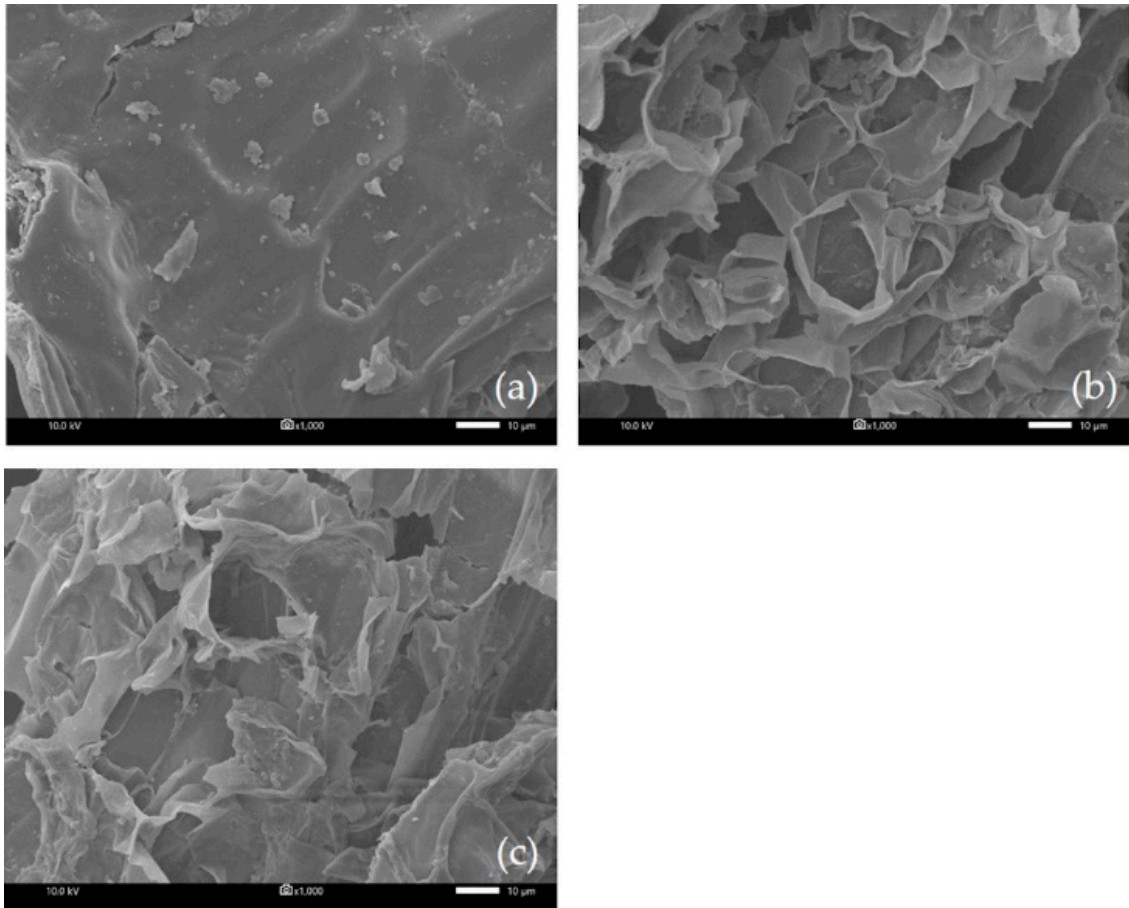

**Figure 6.** SEM image of Siam cardamom leaf sheath (**a**) raw material, (**b**) Siam cardamom treated by MA-SDE, and (**c**) Siam cardamom treated by SDE.

## 4. Conclusions

An optimization procedure using RSM in combination with BBD was successfully applied in order to optimize effective parameters for the performance of MA-SDE. The optimal extraction conditions were as follows: an extraction time of 87.68 min, microwave power of 217.77 W and the material-to-water ratio of 1:13.18 g/mL. Material-to-water ratio had the most impact on the essential oil yield of Siam cardamom, followed by microwave extraction time and microwave power. MA-SDE was found to be effective, enabling a substantial reduction in extraction time, providing an essential oil with yields, 1,8-cineole and oxygenated contents slightly higher than those using traditional SDE. GC–MS results revealed that the essential oil of leaf sheath cardamom contained a large amount of oxygenated monoterpenes. The most abundant substance from the leaf sheath essential oil was 1,8-cineole, similar to those in seeds. However, the other aroma profiling was different. Siam cardamom essential oil obtained by MA-SDE exhibited strong antibacterial effects against the tested Gram-positive bacteria. SEM micrographs confirmed that the microwave technique efficiently supported the release of essential oil by breaking down the cell structure of the Siam cardamom leaf sheath. Therefore, this modified technique MA-SDE appears to be an alternative for the separation of essential oil from Siam cardamom leaf sheath. Leaf sheath of Siam cardamom is an alternative source of the essential oil-rich antibacterial compound, 1,8 cineole.

**Author Contributions:** Conceptualization, P.S. and K.T.; Investigation, P.S., K.B. and K.T.; Software, P.S.; writing—original draft, P.S.; writing—review and editing, P.S., N.W. and K.T.; supervision, N.W, P.S., P.P. and K.T. All authors have read and agreed to the published version of the manuscript.

**Funding:** This research was funded by King Mongkut's University of Technology, North Bangkok (Contract no. KMUTNB-61-NEW-021). Moreover, the second author thanks for the partially support by Naresuan University, Phitsanulok, Thailand (R2562B091).

**Acknowledgments:** We would like to thank the Faculty of Science, Energy and Environment, King Mongkut's University of Technology North Bangkok (Rayong Campus) for the instruments. We also thank Ansaya Choongern and Sasitron Boonprakom for their help in extraction process.

**Conflicts of Interest:** The authors declare no conflict of interest.

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
