# Peer review of "Process Optimization of Microwave Assisted Simultaneous Distillation and Extraction from Siam cardamom using Response Surface Methodology"

_processes, doi:10.3390/pr8040449_

Round 1
Reviewer 1 Report
This interesting and well-performed study deals with the optimization of the extraction process of the EO of Siam cardamom leaf sheath by MA-SDE, in order to maximise the extraction yield, as well as the 1,8-cineole (And oxygenated monoteprenes in general) content. The best time of extraction, solid-to-water ratio and MW power have been evaluated by RSM-BBD and a SEM analysis was performed to assess the effect of the extraction process on the plant matrix. Moreover, the EO antibacterial activity was tested on several bateria strains, in comparison with pure 1,8-cineole as control.
I appreciated the multidisciplinary approach used, which merged the extraction implementation, the chemical analysis and the bioactivity evaluation.
The introduction presented the state of the art and explained the reason for the importance of this study. I liked the materials and methods section, which was very clear and informative. The results are well presented and commented, with the support of a thorough literature overview. The conclusions are well supported by the presented results.
I have some minor comments for the authors (which I also included as comments in the attached pdf):
- The complete botanical name of the analysed species is Amomum krervanh Pierre ex Gagnep.
- The prefixes trans, cis, iso, and para should be written in italics, according to the chemical nomenclature.
- The references (Xue-Feng et al. (2014) reported as ref. 17, Ashokkumar et al. (2019b) reported as ref. 18, and Mahmud (2008) reported as ref. 20)in Table 3 are missing in the bibliography. Moreover, I'd suggest the authors to report in the first column of this table (below the name of the Country of origin of the accessions) the phrase "Reported as:" prior to the botanical name reported, since they are either wrongly reported (Amomum kravanh is a frequent mis-spelling) or mis-matched with another species (Elettaria cardamomum).
- The peak area normalization method does not provide enough precision to report the relative abundances with more than one significant figure (Table 2).
- Moreover, the standard deviations should be reported and a one-way ANOVA should be performed to assess the statistical significance of the compositional differences among the two EO compositions (Table 2).

Author Response
Dear Reviewer 1
Our response was attached.
your sincerely
Panawan

Reviewer 2 Report
The authors of the manuscript " Process Optimization of Microwave Assisted Simultaneous Distillation and Extraction from Siam Cardamom using Response Surface Methodology" described the optimization of the extraction method of the leaf sheath to get the essential oil enriched on 1,8-cineole. Length of the paper is acceptable and data were well-presented in tables and figures. Although some corrections must be included.
The English must be polished specially in the abstract and introduction. For example, there are unnecessary explanation in the abstract (why did the authors use certain methodology?). The authors must expand the explanation in the introduction or discussion. Rewrite the paragraph in page 2 lines 44-48. It is difficult to understand.
Is there any chance that the volatiles will lose due to the use of the rotary evaporator? Be precise and describe the pressure in the rotary evaporator system.
It is well known that microwave can catalyze several chemical reactions. How do the authors ensure that there are no changes in the profile after extraction??? Please expand and discuss
Chemical descriptors must be in italics, as well as m/z.
Describe the name of the plant using the accepted name, check the plant list u other botanical source to describe the name of the plant, and use that through the manuscript
There are a couple typos for example page 6 line 192. Tem??
Overall, the manuscript needs to improve, could be accepted with changes
Author Response
Dear Reviewer 2
Our response was attached.
your sincerely
Panawan

Round 2
Reviewer 2 Report
Now the manuscript looks great for me
Sincerely